# CoTFormer: A Chain-of-Thought Driven Architecture with Budget-Adaptive Computation Cost at Inference

**Amirkeivan Mohtashami**[*]
EPFL

**Matteo Pagliardini**[*]
EPFL

**Martin Jaggi**
EPFL

## Abstract

Scaling language models to larger and deeper sizes has led to significant boosts in performance. Even though the size of these models limits their application in compute-constrained environments, the race to continually develop ever larger and deeper foundational models is underway. At the same time—regardless of the model size—task-specific techniques continue to play a pivotal role in achieving optimal downstream performance. One of these techniques, called Chain-of-Thought (CoT), is particularly interesting since, as we point out in this work, it resembles employing a deeper transformer through re-applying the model multiple times. However, a key subtlety in computing the attention of past tokens differentiates CoT from simply applying the model several times. Based on this insight, we propose CoTFormer, a novel architecture which closely mimics CoT at the token level, allowing us to obtain significantly improved accuracies close to much larger models. While applying CoT introduces additional computation costs, we compensate for it by leveraging CoTFormer's special compatibility with token-wise variable depth. Through a compute adaptive model—which automatically allocates the compute to tokens that need it most—we show that it is possible to reduce the computation cost significantly without any reduction in accuracy, and with further compute cost reductions possible while maintaining a competitive accuracy.

## 1 Introduction

Large foundational models have demonstrated remarkable performance across various tasks, predominantly employing the Transformer architecture (Vaswani et al., 2017). The ability to tackle new tasks in zero-shot or few-shot settings (Brown et al., 2020) has been attributed to emergent properties that become increasingly prominent with model size (Wei et al., 2022a). This observation has sparked a race to build progressively larger models (Brown et al., 2020; OpenAI, 2023; Touvron et al., 2023a;b).

However, despite the evident improvement in performance with size, certain challenges persist even in very large and deep models. One example is their proficiency in mathematical tasks (Cobbe et al., 2021). In response to these challenges, an alternative approach called Chain-of-Thought (CoT) (Wei et al., 2022b) has been proposed, requiring models to think step by step and articulate their thought processes, showing remarkable success (Kojima et al., 2022). In particular, using CoT can improve the general performance of even smaller models Ho et al. (2023); Li et al. (2024).

In this work, we draw attention to the intrinsic connection between constructing deeper Transformers and employing CoT. At a first glance, applying CoT with $n$ thought tokens can resemble an $n$-times deeper Transformer with weight tying implemented on every $n$-th layer (see Figure 1). Such weight tying schemes have been explored in the past (Dehghani et al., 2019). However, in this work, we point out that there is a distinction between CoT and conventional weight tying. More particularly, when applying CoT, the attention mechanism can access previous intermediary tokens whereas in the weight tying such access is not granted.

Based on this observation, we propose CoTFormer, a Transformer that implicitly applies a similar mechanism to CoT. We empirically show that using CoTFormer allows us to obtain much better performances than deeper baseline models. Especially, CoTFormers surpasses existing methods

---

[*]Equal contribution. order is alphabetical.

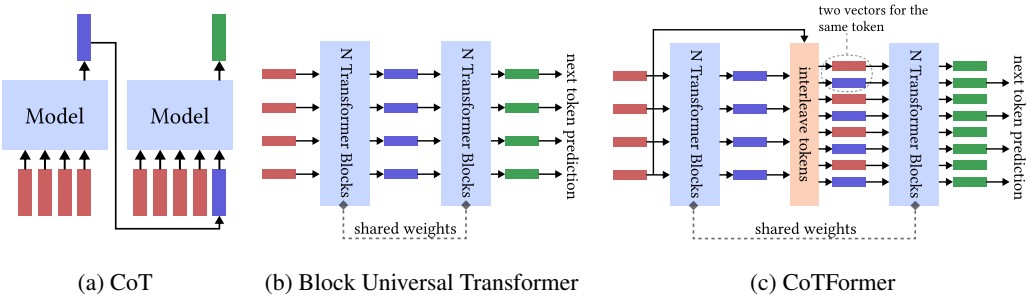

Figure 1: **Block universal transformer vs. CoTFormer vs. Chain-of-Thought (CoT) reasoning.** In **(a)** we represent the chain-of-thought mechanism in which a model is iteratively generating reasoning tokens to help solve downstream applications. Based on existing input red tokens, a next token (blue) is generated and added to the sequence, re-iterating this process yields the green and yellow tokens. Here we emphasize how (i) the last red tokens is "pushed" several times through the model—the yellow token being the red token after three successive applications of the model—and (ii), new (e.g. blue) tokens can attend to previous (e.g. red) tokens, this observation is the basis of CoTFormer. In **(b)** we represent the block-universal transformer which recursively applies the same $N$ transformer blocks to the input tokens. This approach is to be contrasted with the CoTFormer architecture **(c)** which interleaves old and new representations in between each block. In the figure this process is done two times ($n_{\text{repeat}} = 2$), but could be repeated many more times. As in CoT, and unlike block universal transformers, later (e.g. blue) tokens can attend to earlier (e.g. red) tokens.

such as Universal Transformers Dehghani et al. (2019), and pushes the perplexity-compute Pareto frontier forward.

Through asking the model to think step by step, CoT generates a variable amount of intermediary tokens. More complex next token predictions tasks (e.g. an advanced math question) might require to make explicit a greater number of intermediary reasoning steps before reaching an answer. In contrast, tokens which are simpler to predict might not require any intermediary reasoning step at all. This adaptability of CoT to the difficulty of the prediction is remarkable. Indeed, building compute adaptive models has been a long-standing goal, driving the exploration of architectures that can be recurrently applied—controlling the compute cost through deciding the depth of the recursion Banino et al. (2021); Dehghani et al. (2019); Elbayad et al. (2020); Graves (2017); Liu et al. (2020); Tan et al. (2023). However, one challenge those prior methods face is how to allow deeper layers to attend to tokens that have been assigned less depth—e.g. what is the expected interactions between tokens $w_5$ and $w_2$ at depth 3, given that $w_2$ stopped at depth 1? Existing works have proposed possible solutions such as copying the output of the last layer onward. However, these solutions require the model to be able to process the output of different layers at each layer. In contrast, by treating each new application of the model as creating a new token, CoTFormers can completely bypass this problem. Token $w_5$ can simply access all the tokens which have been generated before through the attention mechanism. This makes the CoTFormer a much more natural fit to use in a computation adaptive setting.

In this work, we also propose a new adaptive training method for CoTFormer. We show that using this method yields a model that allows choosing the computation cost at inference time, and navigating the accuracy-computation trade-off without additional training. This is in contrast with most current models that have a fixed computation requirement, preventing them from functioning in more constrained settings. Our model automatically allocates more compute to the tokens that need it while cutting back on others to remain within budget. We observe that, as expected based on our conjecture, the computation cost can be reduced to a certain level with a negligible impact on the accuracy. We also show that, as expected, reducing the computation cost beyond a certain level inevitably reduces the accuracy of the model.

Our main contributions can be summarized as follows:

- Pointing out an important distinction between Chain-of-Thoughts and the recurrent application of a model with weight-tying.

- Proposing CoTFormer which accounts for the aforementioned distinction, and demonstrating its superior performance over other weight-tied deep models (e.g. Universal Transformer (Dehghani et al., 2019)).

- Proposing a training method that allows adjusting the per-token depth distribution at inference, controlling the computation costs while trading compute for accuracy.

## 2 RELATED WORKS

A model usually receives a mix of easy and hard examples which encourages the idea of adapting computation cost based on the input's difficulty. Prior work proposed different approaches to achieve this adaptiveness for various models Bolukbasi et al. (2017); Graves (2017).

These methods usually rely on applying the model multiple times, simulating a deeper model with weight tying. In many aspects, this approach is similar the widely used technique of instructing the model to generate intermediary thoughts before outputting the final answer, called Chain of Thought (CoT) Wei et al. (2022b). Previous work has observed that applying CoT significantly improves performance on various tasks such as mathematical reasoning and its effect on increasing depth has been studied from a theoretical perspective Feng et al. (2023). Furthermore, while Transformers with fixed depth are not Turing complete on their own Merrill & Sabharwal (2023), combining them with the auto-regressive decoding used for generating the chain of thought can make them Turing complete Malach (2023); Merrill & Sabharwal (2024). In this work, while we acknowledge the similarity between CoT and recurrently applying the model, we point out an important difference between these two approaches. Taking this difference into account to mimic the CoT process leads to the development of CoTFormer.

For Transformers, Dehghani et al. (2019) propose Universal Transformers which repeatedly applies a single layer transformer model on the model. A predictor is trained using the ACT method Graves (2017) to decide whether to stop or apply the model again. Due to the instability of ACT and its sensitivity to hyperparameters, Banino et al. (2021) propose PonderNet which weights the predictions at each depth using a probability distribution close to a geometric distribution. This architecture has been extended to cases where the base model has more than one layer. The Block Universal Transformer architecture we explored in this work as a baseline is an example of such architecture while other weight tying arrangements are possible and are explored in Takase & Kiyono (2021).

In these methods, the artificial depth is determined separately for each token. The varying depth between tokens introduces the problem of missing information from tokens that terminated in an early layer when using the attention mechanism in deeper layers. Various approaches have been proposed to address this issue such as copying the latest layer representation forward Liu et al. (2021). In contrast, no such problem exists in CoTFormer since going deeper is equivalent to producing a new implicit thinking token.

Furthermore, the token-based variability of depth makes it challenging to implement batching for these models efficiently. To address a similar challenge when deciding whether to skip blocks of a standard Transformer architecture, Raposo et al. (2024) propose Mixture-of-Depth (MoD) defining a fixed capacity for each block which determines the number of tokens that will go through that block. We use a similar method to allow efficient implementation of our depth adaptive CoTFormer models. However, unlike CoTFormers, MoD uses different weights for each block and therefore does not benefit from the smaller size induced by weight-tying as in CoTFormers. Furthermore, in contrast with CoTFormers which apply a full prefix of blocks, MoDs decide separately whether to use each block or not, preventing early exiting.

Recent work have explored and proposed a variety of alternative architectures which prove to be useful in different scenarios Pagliardini et al. (2024); Wang et al. (2022). Most prominently Mixture-of-Experts (MoEs) have been shown to improve performance of the model in many cases Jiang et al. (2024). For example Sparse Universal Transformers Tan et al. (2023) combine the idea of universal transformer with MoEs, allowing a router to choose a possibly different model every time the input is processed again. In this work we mainly focus our experiments on the Pre-LayerNorm Transformer architecture Xiong et al. (2020), which is currently the most widely used architecture and is the backbone of the state of the art language models Jiang et al. (2023); Touvron et al. (2023c). However, we emphasize that our method uses the Pre-LayerNorm Transformer architecture as a black box and therefore could be directly applied to any of its other variants.

Recent work have also studied explicitly teaching the model to reason by training it on a corpus containing step by step reasoning Nye et al. (2021) and have shown it to be useful. The main obstacle with this approach is the lack of abundant volumes of high quality reasoning data, encouraging recent

work to generate artificial data Ho et al. (2023); Zelikman et al. (2024). Regardless, we believe this approach to be complimentary to CoTFormers. Intuitively, CoTFormers allow re-using basic modules such as extracting information from the context and applying them multiple times. On top of that, the explicit CoT training teaches the model how to reason in a higher level to make rationale arguments.

Finally, while Block Universal Transformers simulate a deeper model, and while alternative proposals such as Pause Tokens simulate a model with increased hidden dimension (i.e. width) Goyal et al. (2024), CoTFormers intuitively facilitates both. Still, width-increasing methods such as Pause Tokens can be combined with CoTFormers.

## 3 CHAIN-OF-THOUGHT AND MODEL DEPTH

Chain-of-Thought involves asking the model to output the solution step-by-step (a process similar to thinking) before outputting the final answer. This process results in the generation of thought tokens in addition to the normal tokens. These thought tokens are generated using auto-regressive decoding. Notably, the whole process of generating thought tokens and finally generating the next normal token is similar to recursively applying the same model multiple times (similar to a Recurrent Neural Network, RNN Rumelhart et al. (1986); see Figure 1a). Consequently, one might be tempted to frame the chain-of-thought process as the utilization of a deeper model with tied weights (see Figure 1b). Indeed, such arrangement resembles a version of Universal Transformers (Dehghani et al., 2019) generalized to allow multi-layer base blocks (instead of only single layer). However, in this work, we point out one critical distinction between the described generalization of universal transformers (which we call Block Universal Transformer), and Chain-of-Thought: **When applying CoT, the generated thought tokens can directly attend to previous thought tokens.**

Having emphasized the above distinction, we propose CoTFormer to closely resemble the CoT process at the token level, taking the highlighted distinction into account.

### 3.1 COTFORMER

Given a context input sequence at depth 0: $\boldsymbol{x}_{1:n_{\text{seq}}}^{(0)} = [\boldsymbol{x}_1^{(0)}, \ldots, \boldsymbol{x}_{n_{\text{seq}}}^{(0)}]$ and a current input token $\boldsymbol{x}_{n_{\text{seq}}+1}^{(0)}$, we describe the process of generating the next token for the Block Universal Transformer and our CoTFormer model. First, let $B^{(i)}(\boldsymbol{x}, \boldsymbol{c})$ be the i-th repeat of a set of $n_{\text{layer}}$ transformer blocks taking as input the token $\boldsymbol{x}$ and being able to attend to the context $\boldsymbol{c}$ through its attention mechanism. One can imagine $\boldsymbol{x}$ being the current token being processed and $\boldsymbol{c}$ being the key/value-cache, as often used during inference. For a typical transformer, generating the output of $B^{(i+1)}$ for token $\boldsymbol{x}_{n_{\text{seq}}}^{(i)}$ can be written as follows:

$$\boldsymbol{x}_{n_{\text{seq}}+1}^{(i+1)} := B^{(i+1)}(\boldsymbol{x}_{n_{\text{seq}}+1}^{(i)}, \boldsymbol{x}_{1:n_{\text{seq}}}^{(i)}). \tag{1}$$

Repeating the above formula $n_{\text{repeat}}$ times with weight tying between the $B^{(i)}, 1 \leq i \leq n_{\text{repeat}}$, yields the Block Universal Transformer:

$$\boldsymbol{x}_{n_{\text{seq}}+1}^{(i+1)} := B(\boldsymbol{x}_{n_{\text{seq}}+1}^{(i)}, \boldsymbol{x}_{1:n_{\text{seq}}}^{(i)}). \tag{2}$$

CoTFormer also use weight tying, but in contrast with the Block Universal Transformer, it provides intermediary representations from previous repeats in the attention. The CoTFormer can be specified as follows:

$$\boldsymbol{x}_{n_{\text{seq}}+1}^{(i+1)} := B(\boldsymbol{x}_{n_{\text{seq}}+1}^{(0)}, [\boldsymbol{x}_{1:n_{\text{seq}}}^{(i)}, \ldots, \boldsymbol{x}_{1:n_{\text{seq}}}^{(i)}]). \tag{3}$$

Figure 1c illustrates the above process. It can be seen that the sequence length grows linearly with the number of repeats. This does not have an effect on memory since the intermediate representations need to be stored in any case, either for the backpropagation during training, or for the KV-cache at inference. However, it may impact the computational cost which we discuss in Section 3.2. We use the notation $n_{\text{layer}} \times n_{\text{repeat}}$ to describe a CoTFormer or Block Universal Transformer with $n_{\text{layer}}$ layers being repeated $n_{\text{repeat}}$ times. Furthermore, we use same position ids for the intermediary representations as the corresponding original token.

### 3.2 COMPARISON WITH BLOCK UNIVERSAL TRANSFORMER

**Experimental setting.** To establish the importance of attending to previous intermediary states, we compare the performance of CoTFormer and Block Universal Transformer on the OpenWeb-Text2 (Gao et al., 2020) dataset; a dataset of websites linked from reddit between 2005 and 2020

Table 1: **Performance of CoTFormer, Block Universal Transformer and Standard Transformers on OpenWebText2**. The mean perplexity of 3 runs is reported with the standard error of the mean in parenthesis. It can be seen that CoTFormers clearly outperforms Block Universal Transformers. The best perplexity for a given $n_{\text{layer}}$x$n_{\text{repeat}}$ combination is marked in bold.

| Model | Base Layers ($n_{\text{layer}}$) | $n_{\text{repeat}}$ | | |
|---|---|---|---|---|
| | | 2 | 3 | 5 |
| Standard | 12 | 28.39 (0.01) | | |
| Block Universal Transformer | 12 | 27.74(0.01) | 27.47(0.01) | 27.15(0.02) |
| CoTFormer | 12 | **27.55(0.02)** | **27.07(0.01)** | **26.64(0.04)** |
| Standard | 24 | 25.93 (0.02) | | |
| Block Universal Transformer | 24 | 25.47(0.01) | 25.19(0.03) | 24.95(0.01) |
| CoTFormer | 24 | **25.28(0.00)** | **24.85(0.04)** | **24.48(0.03)** |
| Standard | 48 | 24.17 (0.00) | | |

initially released under MIT license. We train the models for 40k steps using the AdamW (Loshchilov & Hutter, 2019) optimizer and apply linear warmup of the learning rate for the first 8000 steps. We use the Pre-LayerNorm Transformer Xiong et al. (2020) with 12 heads, hidden dimension 768, sequence length 256, and maximum learning rate 0.001 and feed the data in batches of 128 sequences. We run all our experiments on Nvidia A100 80GB GPUs.

**Perplexity comparison.** The results are reported in Table 1. It can be clearly seen that CoTFormers significantly outperform Block Universal Transformers with the same size and the same number of repeated applications of the model. We emphasize that using CoTFormers **does not** introduce an overhead in terms of memory. This is because the storage of intermediary tokens is needed given the need for the KV cache even when using Block Universal Transformers.

**Compute cost comparison.** As such, the only downside of using CoTFormers instead of a Block Universal Transformer is the growth in the computation cost of the attention layer. This growth occurs because when using CoTFormers, the outputs of all previous repeats are accessible. Therefore, given the quadratic cost of the attention, one might expect the cost of CoTFormer to grow quadratically with number of repeats. However, for current models, the main bottleneck in computation when processing average length sequences is the feed-forward network in each block, not the attention layer Ganesh et al. (2021); Tay et al. (2022). It is only for very long sequences that the attention layer becomes a bottleneck. Therefore, the growth in computation cost is actually much less noticeable. At the same time, using CoTFormer comes with significant boost in accuracy. The same pattern can be observed in Figure 3 which shows the number of multiply-accumulate operations needed to process different sequence lengths by a block universal transformer with $n_{\text{repeat}} = 5$ and a CoTFormer with $n_{\text{repeat}} = 3$ which obtains a similar accuracy (on sequences of length 256). It can be seen that even for sequences as long as 8192, the 12x3 CoTFormer's cost remains below that of 12x5 Block Universal Transformer.

The demonstrated trade-off is further depicted in Figure 2 which shows the perplexity against compute cost of processing a sequence with 256 tokens for CoTFormers and Block Universal Transformers with $n_{\text{layer}} = 12$ and $n_{\text{layer}} = 24$. At both scales, it can be clearly seen that while CoTFormers with the same number of repeats are more costly, they come with significant improvement in accuracy which overall puts them in the front of the Pareto frontier. Furthermore, the performance gap widens as the number of repeats increases, suggesting better scaling properties of CoTFormers. The above results clearly demonstrate the effectiveness of CoTFormers over Block Universal Transformers.

### 3.3 ARCHITECTURE TWEAKS & LN-COTFORMER

The previous section introduced as little innovations as possible to clearly demonstrate that the improved performances are due to the built-in CoT mechanism, and not to some other tweak. Having established the better performance of CoTFormer, we now introduce several modifications which we found further improved the results.

**Reserving Beginning and Final Layers.** In order to allow the model to operate on an intermediary space which is not necessarily the same as the word embedding space, we propose separating the first and last few layers from repeats. In this scenario, the model will first execute $n_{\text{begin}}$ layers, followed by multiple passes through $n_{\text{middle}}$ layers. Finally the output is generated from the final pass of each token by passing it through the last $n_{\text{end}}$ layers.

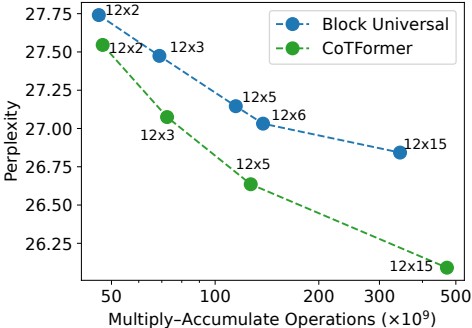

(a) $n_{\text{layer}} = 12$ (x-axis is in log scale)

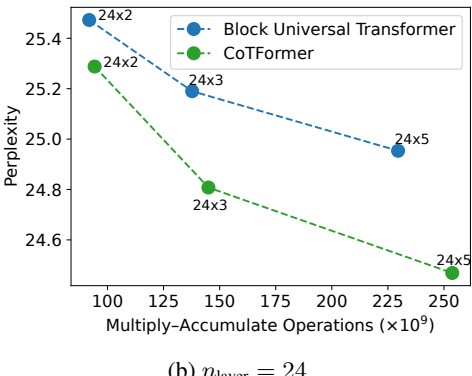

(b) $n_{\text{layer}} = 24$

Figure 2: **Comparison of Block Universal Transformer and CoTFormer in terms of accuracy-computation tradeoff.** It can be clearly seen that at both $n_{\text{layer}} = 12$ and $n_{\text{layer}} = 24$, CoTFormers are closer to the Pareto frontier. The gap widens with larger number of repeats, suggesting better scaling properties of CoTFormers. The x-axis shows the number of operations for processing a sequence of length 256.

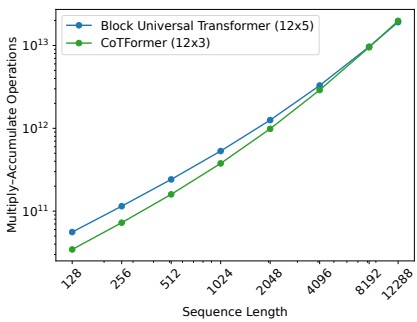

Figure 3: **CoTFormer is less compute intensive than a Block Universal Transformer of comparable performance.** Comparing a 12 layers CoT-Former with 3 repeats (12x3) and a 12 layer Block Universal Transformer with 5 repeats (12x5) in terms of computation cost. The CoTFormer's accuracy is better than the Block Universal Transformer (see Figure 2). Despite the increase in context length when processing the input with CoTFormer, the computational cost of CoTFormer remains below the Block Universal Transformer for sequence lengths as high as 8192.

Table 2: **Ablation study for the architecture tweaks discussed in Section 3.3.** The final architecture with $n_{\text{repeat}} = 5$ obtains lower perplexities than a 48 layers standard transformer which has double its size.

| Model | $n_{\text{layer}} \text{x} n_{\text{repeat}}$ | Perplexity |
|---|---|---|
| Standard | 48x1 | 24.17(0.00) |
| CoTFormer | 24x5 | 24.48(0.03) |
| + Reserved Layers | 2→21x5→1 | 24.51(0.01) |
| + Layer Norm | 2→21x5→1 | **24.11(0.03)** |

**Layer Norm After Each Repeat.** Given the internal residual connections of the model, we conjecture that it is important to maintain a consistent input's scale. Therefore we additionally inject a layer norm at the end of each repeated pass, similar to the final layer norm applied in the standard architecture before predicting the next token.

The clear positive effect of the above tweaks on performance can be seen in the ablation study in Table 2. In the case of reserved beginning and final layers, note that while the accuracy does not improve, the computation cost decreases since the total number of layers is kept fixed at 24. Most noticeably, after applying these changes, the performance of a CoTFormer with 24 layers and 5 repeats, surpasses the performance of a standard 48 layer Transformer. We present similar results for downstream tasks in Appendix B. We call the final resulting architecture LN-CoTFormer. We note that while LN-CoTFormer's final performance is better than CoTFormer, we observed some benign spikes in the loss during training. Though the model quickly recovers without intervention from these spikes, this might suggest CoTFormer are more stable to train than LN-CoTFormers. Still, we focus on using LN-CoTFormers when building our adaptive model in the next section.

## 4 TOKEN-WISE ADAPTIVE REPEATS

The standard CoTFormer has the advantage of obtaining better performance with smaller models which is useful in memory-constrained environments such as mobile phones. Moreover, the recurrent application of the small model also opens up the direction of varying the number of times the model is applied, i.e. the number of repeats, on a more granular level. In particular, the intuition that the difficulty of predicting the next token varies over the sequence, encourages that dynamically varying the number of repeats on a token level based on the context can yield computational savings.

Prior work, in particular universal transformers, also aim to create such adaptive models that use a different number of repeats based on the difficulty of the current token. This is done through a halting module which is called at the end of each repeat to decide whether the current state should be used as the output (i.e. halt) or to continue with another pass through the small model. However, two challenges remain persistent:

- **Attending to Previously-Halted Tokens in Later Layers**: If a token decides to halt early, subsequent repeats after the one where the token halts still need to have a representation of the token available in the attention layer to allow tokens that are still processing access the already halted token. Prior work have suggested approximating this representation by copying the last output for the token forward. In contrast CoTFormer does not face this challenge. Since the model can attend to each token's representation after each of previous repeats, a halted token is already represented when invoking the attention mechanism. As such, CoTFormer adapts much more naturally to the adaptive depth setting.

- **Efficient Batch Processing**: Since the decision of halting or continuing happens on a token level, sequences of the same length may end up with different number of tokens in the subsequent repeats. As a result, efficient processing of batches of multiple sequences becomes challenging. Therefore, in this work, we propose a different approach where a certain capacity is assigned to each iteration of processing the sequence using the small model, and the most eager tokens are assigned to pass through that model again. Our approach is similar to the method proposed by Raposo et al. (2024) to train Mixture of Depth (MoD) models in some aspects, namely assigning capacities for each iteration instead of for tokens, but deviates from MoD's training method in other aspects, such as randomized capacities, as detailed in the next subsection.

In addition to addressing the above challenges, we also aim to build a model that can work under different constraints. In particular, we aim to offer the flexibility to choose the compute budget during inference, with more compute yielding a better accuracy. Therefore, we randomly pick the compute budget at each iteration instead of fixing it in advance, allowing the model to adapt to different constraints. Similar approaches have been used to build models with varying width or rank Mohtashami et al. (2022); Yu et al. (2018). We now explain the details of our method.

### 4.1 MIXTURE OF REPEATS

We assume all tokens go at least one time through the model. We now use $n_{\text{repeat}}$ to refer to the maximum number of times a token can go through the model. For each of the passes 2 to $n_{\text{repeat}}$ we train a separate embedding vector, e.g. $e^{(i)}$ for the $i + 1$-th pass, and use the dot product between this vector and the current representation of the token. In particular, if the output after the $i$-th pass for $j$-th token is denoted by $x_j^{(i)}$, we compute the score $s_j^{(i)} := \sigma(e^{(i)\top} x_j^{(i)})$ to determine whether the $j$-th token should pass for the $i + 1$-th time through the model. We interchangeably use the terms router to refer to this mechanism. The router weights (the vectors $e^{(i)}$) are trained alongside other parameters of the model.

To determine which tokens pass through the next repeat of the model, we sort the tokens based on their score as defined above and pick the top $k$ where $k$ is chosen based on the capacity level for this repeat: $c_i$. In particular, if we denote the sequence length by $n_{\text{seq}}$, we will use $k := \lfloor c_i \times n_{\text{seq}} \rfloor$.

Let us denote the output of the model on the input $x$ by $B(\boldsymbol{x})$. We use an interpolation between the previous pass's output and the new output of model to get the output of this pass. In particular, we use $\boldsymbol{x}^{(i+1)} := (1 - s_i) \cdot x_i + s_i \cdot B(\boldsymbol{x}^{(i)})$. This interpolation plays two roles. First of all, the gradient needed to train the embeddings $e^{(i)}$ is obtained only based on this interpolation since the process of token selection is not differentiable. More importantly, it provides a way to the model to ensure that increasing capacity will not hurt the performance. For example, if we set the capacity of a repeat to 1,

even tokens with very low scores will be selected. However, a low score indicates that such additional processing of these tokens might adversely affect the accuracy of the prediction. As a result of this interpolation, the representations of such tokens remain unchanged.

Finally, instead of using fixed capacities $c_i$, we sample them at random for each batch. More particularly, we assign a capacity of 1 to the first repeat and sample $n_{\text{repeat}} - 1$ numbers and sort them in decreasing order to get the capacity for other repeats. This random sampling has two key effects. First of all, intuitively, sampling allows tokens to explore being passed through deeper layers. Otherwise, only tokens with a high score that were selected for a pass would affect the gradient. Therefore, the update for router weights would only take into account those high scoring tokens. As a result low scoring tokens will continue to be excluded. This challenge of exploration vs exploitation arises from simultaneous training of the router weights and the model parameters. The second effect of the sampling is ensuring the model functions with different capacity factors. This allows adjusting the capacity factors at inference, which in turn allows customizing the computation budget without unreasonable losses in accuracy.

## 4.2    ADAPTIVE ARCHITECTURE

We mainly use a LN-CoTFormer to build the adaptive model which has the same architecture as Section 3.2 except for the architecture tweaks discussed in Section 3.3. In particular, we fix (meaning we do not repeat) the first 2 layers and the last layer. Additionally, we train our models for 60k steps instead of 40k steps. Finally, we introduce a depth embedding to the model.

**Depth Embedding.** In order to allow the model recognize how many number of passes it has done, we add a depth embedding at the start of each pass. For this embedding we learn a single vector $e^{(\text{depth})}$ and add $(n_{\text{repeat}} - i) \cdot e^{(\text{depth})}$ as the embedding for the $i$-th repeat. Intuitively, this should condition the model based on the maximum number of repeats it has left. We investigate the effect of this embedding in Appendix Table 4. While the performance is similar on fixed number of repeats, a noticeable boost is observed in the adaptive case.

## 4.3    RESULTS

By design, an adaptive model's performance depends on the amount of compute it is allocated. In order to measure the performance of the model for different computes, we compare two methods. The first is to activate a prefix of repeats, and the second is to rely on the router weights learned by the model. For the second approach, we compute the ratio of tokens that enter each repeat on a subset of the training set and use the obtained ratios as the capacity factors. Alternatively, one could directly threshold the router weight without needing such statistic measurement. However, we chose the former approach for simplicity and maintaining the ability of batch inference.

In Figure 4 we plot the accuracy against the number of multiply accumulate operations. We vary the router threshold to move between compute budgets. In order to show the effectiveness of the router in allocating compute to tokens, we compare the results with the alternative method of running all tokens at inference time on a smaller number of repeats to reduce cost. The results clearly show that relying on the learned router weights provides a far more effective way of allocating compute and manages the accuracy-compute trade-off more efficiently. As a result, the computation cost can be significantly reduced without noticeable loss of accuracy. Further reduction of computation cost is also possible in exchange for reasonable accuracy losses, allowing us to traverse the accuracy-compute trade-off at inference time which is not possible with the standard models.

We also report the results for an adaptive Block Universal Transformer. While alternative methods for adaptive training of Universal Transformers have been proposed in the literature, we could not obtain a better performance using those methods in small scale experiments. We provide additional details in Appendix D and continue with reporting the results of training a Block Universal Transformer using Mixture of Repeats. Following previous work, for already halted tokens, we copy their last representation forward.

Similar to our previous results, we observe that CoTFormer outperforms Block Universal Transformer when enough compute is available. However, when moving to the lowest computation budget, in particular when no repeat is allowed, the adaptive Block Universal Transformer outperforms the adaptive CoTFormer. Intuitively, this can be because CoTFormer learns to better utilize the additional number of repeats to obtain better performance but has to sacrifice some performance when the number of repeats remains low.

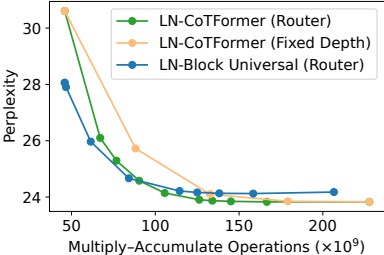

Figure 4: **Perplexity for different amount of compute budgets chosen at inference.** The adaptive CoTFormer can adapt to different budgets, reducing compute in exchange for reasonable loss in accuracy. Furthermore, using the router weights to allocate the available compute (Router) is much more effective than fixing the depth at inference time to a smaller value in order to reduce computation cost (Fixed Depth).

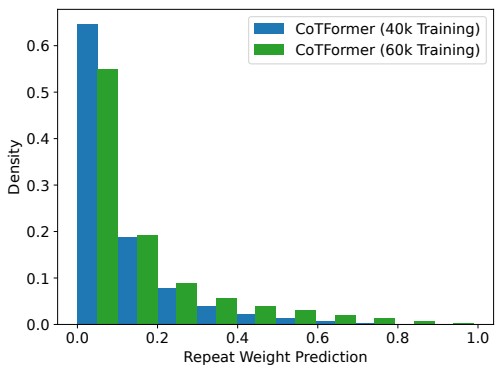

Figure 5: **Distribution of router weights for the last repeat for different number of training steps.** It can be seen that when training longer, the model learns more to use the deepest repeat, leading to higher router weights.

## 5 DISCUSSION AND FUTURE WORK

**Training of Deeper Layers.** While the above performance is remarkable, we can observe a gap between an adaptive CoTFormer and a non-adaptive CoTFormer trained with exactly 5 repeats even when the adaptive variant is allowed the same amount of compute at inference. For example, after 60k steps, the former reaches perplexity 23.83 while the latter achieves 23.19. One possible reason is the reduced amount of gradient information reaching higher number of repeats in the adaptive training since a good portion of tokens will halt before reaching those repeats. As such, we conjecture that the adaptive CoTFormer would benefit more from longer training. We verify this in Figure 5 where we plot the ratio of different values of router weights for the final repeat when the model is trained for 40k steps and compare it with training for 60k steps. We can clearly see that the model starts to favor using the final repeat more when the model is trained for longer. We note that training time of an adaptive model is *significantly lower* than training directly at the maximum number of repeats. For example, when training the model with fixed 5 repeats, the training takes roughly around 1.5x longer.

**Alternative Sampling Methods.** In addition to longer training, we conjecture that the sampling plays an important role in the quality of the final model. While we tried some alternative sampling methods, we could not find a method that performs better than randomly picking and sorting as described in Section 4.1. Still, we expect better methods to exist and leave exploration of such methods as a direction for future work.

**Comparison with Standard Transformer.** Currently, getting the same performance as a deeper standard transformer model, requires more number of repeats than the depth difference factor. For example, to get better performance than a 48 layer model using a 24 layer model, 5 repeats is needed whereas optimally we would need 2. As shown in Table 1, Using CoTFormer instead of Block Universal Transformer, significantly reduces this gap while at the same time maintaining alternative advantages such as smaller model size and the adaptivity of compute at inference. Still, reducing this gap further remains an important direction for future work.

**Efficient Implementation.** Since CoTFormer introduces additional tokens to the sequence, the attention module's implementation, in particular the causal mask, needs to be adapted. In this work, we rely on a simple implementation (using non-causal version of Flash Attention Dao et al. (2022)) which leaves room for improvement. In particular, low-level kernels such as Pagliardini et al. (2023) can be directly used to improve the speed of CoTFormer's implementation.

## 6 CONCLUSION

In this work, we point out an often overlooked distinction between chain of thought and iteratively applying the model. By taking this distinction into account we develop CoTFormer and show its superior performance to Block Universal Transformers. Most noticeably we propose additional small tweaks in the architecture, allowing a CoTFormer to obtain the same accuracy as a standard

Transformer that has double its size. Moreover, we propose an adaptive training method and showed it enables adjusting the compute budget at inference in exchange with reasonable impact on accuracy. Unlike prior methods, our method does not introduce sensitive additional hyperparameters and allows for stable training. Finally, we discuss different avenues to improve the results, particularly in adaptive setting, in future work.

## ACKNOWLEDGEMENT

This project was supported by SNSF grant number 200020_200342.

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

## A    CODE

Our implementations for all experiments is available at https://github.com/epfml/cotformer.

## B    DOWNSTREAM TASKS

We evaluate the zero-shot performance of the models we trained on OpenWebText2 on several downstream tasks, and present the results in Table 3.

| Model | MMLU | ARC | Hellaswag | PIQA | Average |
|---|---|---|---|---|---|
| Standard Transfromer (24) | 25.83 | 29.34 | 27.41 | 58.54 | 35.28 |
| Block Universal Transformer (24x2) | 25.64 | 29.5 | 27.38 | 59.09 | 35.4 |
| Block Universal Transformer (24x5) | 26.07 | 30.25 | 27.73 | 58.27 | 35.58 |
| CoTFormer (2->21x5->1) | 25.98 | 29.89 | **28.2** | 59.3 | **35.84** |
| CoTFormer (24x5) | 25.99 | 29.95 | 27.74 | **59.47** | 35.79 |
| CoTFormer (24x2) | **26.22** | **30.72** | 27.26 | 58.65 | 35.71 |
| Standard Transformer (48) | 26.11 | 29.31 | 27.93 | 60.28 | 35.91 |

Table 3: Accuracy (Noramlized by Sequnece Length, Ignoring Space). The model's result with best performance between Block Universal Transformer and CoTFormers in each task is shown in bold.

As expected, we can observe patterns similar to the perplexity results presented in Section 3.2, with CoTFormer outperforming Block Universal Transformers. We emphasize that these results should be interpreted only in comparison with each other as obtaining good downstream tasks performance requires very long training which is not possible due to computational budge limitations.

## C    EFFECT OF DEPTH EMBEDDING

Table 4: Effect of Depth Embedding on fixed and adaptive number of repeats. The performance is similar on fixed repeats but is noticeably improved in the adaptive case.

| Model | Adaptive | $n_{\text{repeat}} = 5$ |
|---|---|---|
| LN-CoTFormer | 25.08(0.03) | 24.11(0.03) |
| + Depth Embedding | 24.94(0.01) | 24.17(0.08) |

## D   ALTERNATIVE ADAPTIVE TRAINING METHODS

We experimented with Stick Breaking method proposed by Tan et al. (2023) as well as the halting mechanism in PonderNet Banino et al. (2021).

As acknowledged in the same work, we found training with PonderNet mechanism to be challenging and sensitive to the choice of hyperparameter, specifically the weight of the KL divergence. We tried tuning this parameter and report the best result in Table 5.

When using Stick Breaking Halting, we observed that the model tends to be very conservative and as a result the average depth remains too low.

We compare the results of training block universal transformer for 10k iterations in Table 5. We observed better final perplexity with our method (Mixture of Repeats) than the other two methods. Due to computational limits, we could not perform longer experiments but decided to use our method given the more stable and less sensitive training dynamics as well as the better performance.

Table 5: Comparison of Mixture of Repeats with Previous Mehtods.

| Method | Perplexity |
|---|---|
| Stick Breaking | 33.82 |
| PonderNet ($\lambda_p = 0.4$) | 41.37 |
| Mixture of Repeats | **33.08** |

## E   COMPARISON WITH PAUSE TOKENS

In Goyal et al. (2024), the authors show adding a number of virtual tokens, called pause tokens, after each original token in the input, leads to improvements in perplexity. While CoTFormer also adds additional tokens after each original token, these new tokens are not just placeholder tokens. Instead, they are intermediate representations of the model and each subsequent token is created by passing the last token through the model again. To demonstrate that this is important for the performance of CoTFormer, in Table 6, we compare a model with 4 pause tokens and CoTFormer with 5 repeats. It can be clearly observed that CoTFormer significantly outperforms pause tokens.

Table 6: **Comparing the performance of CoTFormer and pause tokens (Goyal et al., 2024)**. Adding pause tokens improves perplexity but it is still heavily outperformed by CoTFormer that uses the output of previous repeat for the subsequent processing by the model.

| Model | Base Layers ($n_\text{layer}$) | $n_\text{repeat} = 5$ |
|---|---|---|
| Standard | 24 | 25.93 (0.02) |
| Pause tokens (Goyal et al., 2024) | 24 | 25.05(0.03) |
| Block Universal Transformer | 24 | 24.95(0.01) |
| CoTFormer | 24 | **24.48(0.03)** |
| Standard | 48 | 24.17 (0.00) |

## F   ATTENTION PATTERNS

In this section, we present some of the attention patterns we observed in a 24 layer CoTFormer with 5 repeats. While, a thorough discussion around understanding how these models operate is outside the scope of the current work, we hope these results encourage such investigations.

In particular, we plot the average attention pattern of the last token in the last repeat in Figure 6. We observe interesting patterns. In particular, we notice heads that attend to different repeats for the current or recent tokens. Moreover, we observe heads that attend to tokens generated during a specific repeat. This is especially common for the first and last repeat though some heads also focus on other

repeats. These patterns suggest that the model does not only rely on having access to intermediate representations of the current token but also uses intermediate representations of the previous tokens.

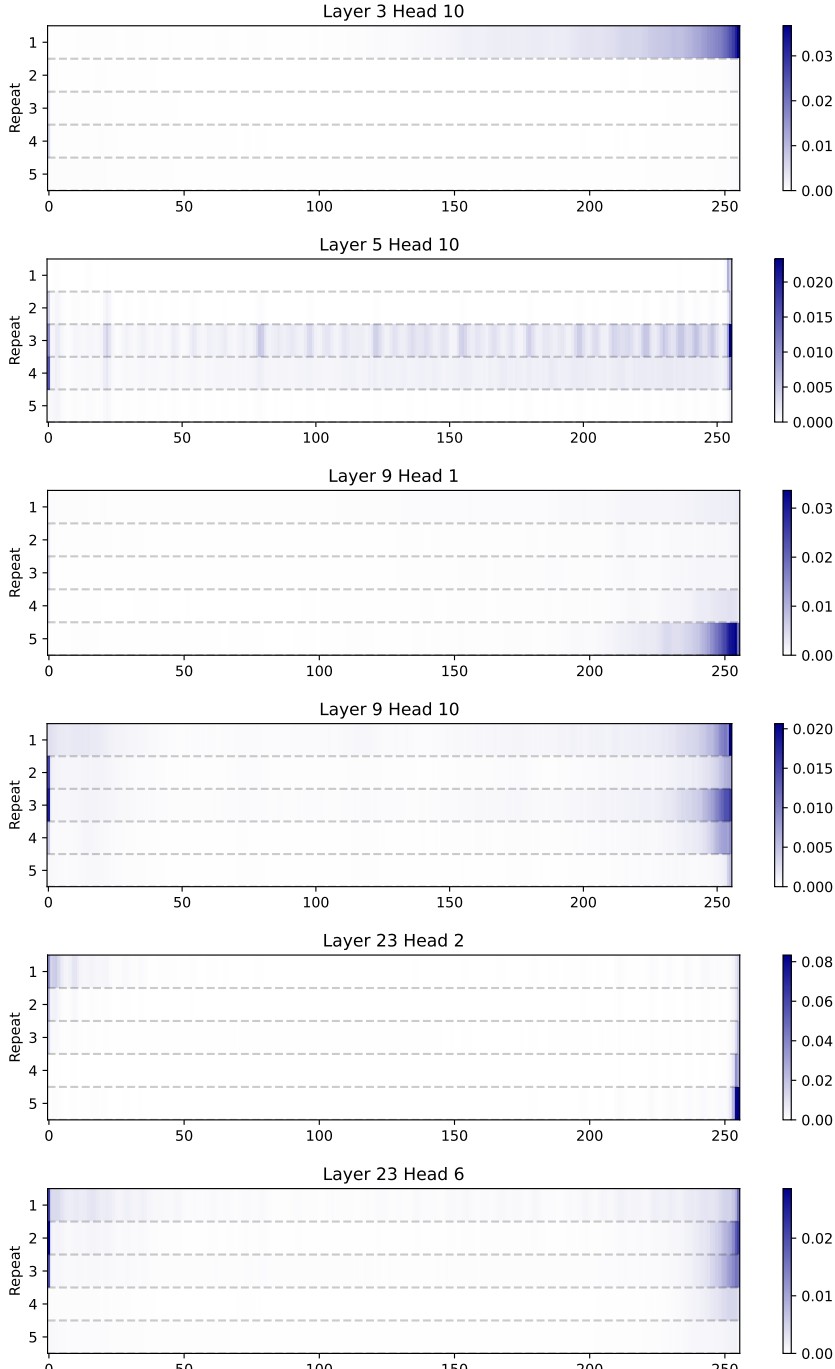

Figure 6: **Attention Patterns of a 24 layer CoTFormer with** $n_{\mathbf{repeat}} = 5$**.** Each figure shows the average of attention scores over validation data from the last repeat of the last token in sequence to all other token-repeat pairs. The x-axis shows the token index whereas the y-axis shows the repeat that the token belongs to. The intensity of the color shows how high the attention score to a specific token at a specific repeat has been for the head being considered (averaged over validation data).

## G    EFFECT OF SEQUENCE LENGTH AND WIDTH

For the main experiments in the paper we focus on models with 768 hidden dimension and 256 sequence length. Here, we also present results for 1024 hidden dimension Block Universal Transformer and CoTFormer and also compare these models with 512 seqeunce length (with 768 hidden dimension). The results are shown in Table 7 and clearly demonstrate the benefits of CoTFormer over Block Universal Transformers persist on different widths and sequence lengths. All models have 12 layers and use $n_{\text{repeat}} = 5$.

Table 7: **Comparing CoTFormer and Block Universal Transformer at different width and different sequence lengths**.

| Model | Base Layers ($n_{\text{layer}}$) | Hidden Dimension | Sequence Length | $n_{\text{repeat}} = 5$ |
|---|---|---|---|---|
| Block Universal Transformer | 12 | 768 | 256 | 27.15(0.02) |
| CoTFormer | 12 | 768 | 256 | 26.64(0.04) |
| Block Universal Transformer | 12 | 768 | 512 | 21.53 |
| CoTFormer | 12 | 768 | 512 | 21.06 |
| Block Universal Transformer | 12 | 1024 | 256 | 24.89 |
| CoTFormer | 12 | 1024 | 256 | 24.56 |

