# OpenReview forum: "CoTFormer: A Chain of Thought Driven Architecture with Budget-Adaptive Computation Cost at Inference"
_ICLR.cc/2025/Conference — ICLR 2025 Poster_

### Official Review · Reviewer_jCWB · 2024-10-17

**Soundness:** 2
**Presentation:** 3
**Contribution:** 3
**Rating:** 6
**Confidence:** 3

**Summary:**

The paper presents CoTFormer, a novel Transformer architecture that leverages the Chain-of-Thought mechanism to enhance model performance while allowing for budget-adaptive computation at inference. CoTFormer enables intermediate tokens to be accessible, improving accuracy without significantly increasing computational costs. The authors further propose an adaptive training method that dynamically allocates computational resources based on the needs of individual tokens. Empirical results demonstrate that CoTFormer outperforms existing models, such as the Block Universal Transformer, while maintaining a smaller model size.

(Note: Thank the authors for their clarification; that addressed some of my concerns. I've adjusted the score accordingly)

**Strengths:**

1. The architecture is novel and the authors made smart observations with respect to the CoT can access previous tokens.
2. The problem is very practical and of high interest to the community, especially with limited computation resources
3. The paper is overall well-written and organized.

**Weaknesses:**

1. The paper could have a more detailed discussion on the scalability of the architecture, with respect to larger models and higher sequence lengths, since the paper discusses that the attention computation is not the bottleneck.
2. The paper studies the performance of CoTFormer on a particular dataset; would be interesting to see the performance on other datasets
3. The paper could have benefited from a more thorough theoretical analysis of COTFormer, especially with the number of repeats compared to the block universal transformer

**Questions:**

1. Is the compute budget a hyperparameter to tune to achieve an optimal balance between accuracy and computation?

---

> ### Author Response · Authors · 2024-11-18
>
> Dear Reviewer,
>
> We provide the following answers to your questions:
>
> 1. We run our experiments on both 12 layer and 24 layer CoTFormers and compare with up to 48 layer Transformers to showcase consistency of our results in different scales. However, scaling up the model further requires much higher computation power. In the revision of our paper, we provide results for 512 sequence length in the appendix which shows the benefits of CoTFormer persists at these lengths as well.
>
> 2. In order to limit the number of experiments and compute costs, we focus on the OpenWebText2 dataset, we emphasize that this is a fairly generic and large dataset and is not task specific similar to the datasets used for state of the art pre-training.
>
> 3. We are not sure what kind of theoretical analysis is requested here. The goal of this paper is to clarify a clear advantage of CoTFormer over Block Universal due to access through attention which has not been pointed out in prior work. This is evidenced by empirical results. Any theoretical analysis remains far outside the scope of this work.
>
> 4. In the adaptive CoTFormer, the compute budget can be decided **at inference** and can be tuned depending on the cost of compute for the user. Of course, more compute usually leads to better accuracy (as shown in Fig. 4).
>
> We hope the above answers adequately answer your concerns. However, looking at your review, we do not understand why you are suggesting a rejection of our work as most of the raised points are not major objections. If our answers have adequately answered your concerns, please consider raising your score or let us know any major objections that compel you to suggest a rejection so we can also answer them.

---

### Official Review · Reviewer_rjXu · 2024-10-29

**Soundness:** 2
**Presentation:** 2
**Contribution:** 2
**Rating:** 6
**Confidence:** 3

**Summary:**

The paper proposes a new model architecture called CoTFormer that improves the Block Universal Transformer by providing intermediary representations from previous repeats in the attention. Besides CoTFormer architecture, the paper also proposes a training approach called Mixture of Repeats that varies the number of model passes for individual tokens based on their difficulty. Results show that CoTFormer substantially improves accuracy and inference computation efficiency over Block Universal Transformer.

**Strengths:**

1. The CoTFormer architecture and Mixture-of-Repeats approach effectively improve the performance and efficiency of the Universal Transformer.
2. The evaluation of downstream tasks illustrates the CoTFormer's potential to surpass the standard Transformer.

**Weaknesses:**

1. Possible misuse of technical terms: Chain-of-thought is a prompting technique. The process illustrated by Figure 1 (a) is called auto-regressive, which is orthogonal to CoT. Could the authors clarify how the CoTFormer model architecture model relates to the CoT prompting? Could CoT prompt be applied to CoTFormer model?
2. The model architecture is not clearly explained. Specifically, the meaning of different colors in Figure 1 is vague. Why are there no yellow tokens in Figure 1 (b) and (c)? The figure can be more clear if the caption explains the reason for the absence of yellow tokens and the meaning of different numbers of tokens.

**Questions:**

1. Figure 2 shows the inference FLOPs vs. Perplexity. However, it cannot suggest better "scaling properties of CoTFormers" (quote Line 257 of the paper) because scaling properties should be suggested by the training FLOPs vs Perplexity following Kaplan et al.[1]. Could you provide the training FLOPs vs. Perplexity plot for Figure 2?
2. Could you add the standard Transformer to Figure 2?
3. The paper claims that "The growth in computation cost is actually much less noticeable". Could you provide the real measurement of computation cost in terms of memory footprint (Figures 2 and 3 only show FLOPs)?

**After discussion period, questions were addressed by the authors:

Answer 1: Keep all the other factors constant, the scaling behavior with respect to the training FLOPs still holds.

Answer 2: The accuracy of the standard Transformer in Table 1 can indicate the distance between the CoTFormer and the standard Transformer. Therefore, it is necessary to add the standard Transformer to Figure 2.

Answer 3: Theoretically, the memory footprint is the same for CoTFormer and Block Universal.


[1] Kaplan, Jared, Sam McCandlish, Tom Henighan, Tom B. Brown, Benjamin Chess, Rewon Child, Scott Gray, Alec Radford, Jeffrey Wu, and Dario Amodei. "Scaling laws for neural language models." *arXiv preprint arXiv:2001.08361* (2020).

---

> ### Author Response · Authors · 2024-11-18
>
> Dear Reviewer,
>
> While we appreciate your consideration of our work, upon reading your review, we do not understand your justification behind rejecting our work with such high confidence. We hope the following comments clarify our contributions further and earnestly ask you to consider raising your score or provide reasons that have so strongly compelled you to reject our work.
>
> 1. Chain of thought is a prompting technique that relies on auto-regressive decoding. Figure 1 is demonstrating how each method works, comparing what happens in CoTFormer with what happens when a Chain of thought technique is used on a normal Transformer. The goal of this figure is to demonstrate the difference between generating CoT using auto-regressive decoding (Figure 1a) and repeatedly applying the same model (block universal, Figure 1b). In the former, a token is fed through the model to generate a new token, which is appended to the sequence, and itself fed again to the model, generating a new token, etc. This is conceptually similar to feeding the same token through multiple repetitions of the model. In the latter approach, each token is fed to multiple repetition of the model, yet—unlike in the former approach—intermediary representation of a given token cannot attend to past representations of that token. This observation led us to the design of CoTFormer. The colors are meant to distinguish tokens based on how many steps were used to generate them. For chain of thought (Figure 1a), the figure shows a process of generating an answer after 3 steps of decoding with the model whereas for block universal and CoTFormer, the figure only shows 2 repeats. That is why there are no yellow colors for CoTFormer and Block Universal. We understand that this can be slightly confusing and will remove the third step from CoT so all the figures correspond to 2 steps of using the model. That being said, given that this is the only weakness mentioned in your review, we strongly object that this would be grounds for rejection.
>
> 3. Regarding training vs inference FLOPs, we note that we are plotting against the FLOPs for a forward pass of sequence length 256.  The backward pass’s FLOPs is usually within a constant factor of the forward pass. As such—given we are using a similar number of training iterations, batch sizes, etc. for all methods—training FLOPs and inference FLOPs are not that different. Note that we are not plotting against single token decoding FLOPs. We will add this remark to our next revision. Let us know if more clarifications are needed on this point.
>
> 4. The goal of this work is to point out the difference between doing CoT using auto-regressive decoding, and simply applying the model multiple times (block universal transformer). We do not claim that we are doing better than a larger standard Transformer and clearly discuss the benefits (such as adaptive compute) and limitations (needing 5 repeats to match 48 layer Transformer) in the paper (Section 5). However, compared with architectures that rely on repeatedly applying the same model, CoTFormer still improves significantly over block universal transformers due to attention’s access to intermediary representations. To draw a parallel with another recent line of research, it is known that dense transformer models are better than Mixture-of-Expert (MoE) models with similar number of total parameters, yet MoEs have other benefits that make them desirable leading to them being widely adapted by the community.
>
> 5. Please note that the referred sentence about the growth of computation cost of attention being less noticeable is about much larger models that have large feed forward layers. Training these models is not possible for us due to resource limitations. However, we point out that when using flash attention, the attention’s memory cost increases linearly (not quadratically) with length. Therefore, theoretically when factoring in the KV cache, the memory requirements of Block Universal and CoTFormer are the same.
>
> We believe we have addressed (i) the misunderstanding regarding the link between our method and CoT, as well as (ii) the performance of our method in terms of training FLOPs, and  finally (iii) compared the memory footprint of our method with our block universal transformer baseline. As those fully cover the concerns raised in your review, we hope you will either raise your score or provide further clarifications justifying your score.

---

> > ### Comment · Reviewer_rjXu · 2024-12-01
> >
> > Thank the authors for the response. I agree the backward FLOPs are about twice the inference FLOPs. However, training FLOPs = backward FLOPs * the number of tokens trained. Current inference FLOPs vs. Perplexity certainly demonstrates some scaling properties. I would adjust my claim to be that training FLOPs vs. Perplexity could suggest more scaling properties. I agree that optimizing block universal transformers can be justified as an interesting research problem. However, the distance between CoTFormer and the standard Transformer should at least be demonstrated from an "upper-bound" analysis perspective. I have raised the score but I hope these clarifications make sense to the authors.

---

> > > ### Author Response · Authors · 2024-12-01
> > >
> > > Dear Reviewer,
> > >
> > > Thank you for the clarification and updating your score.
> > >
> > > Regarding the training FLOPs, note that we train all models for the same number of steps, the same batch size, and the same sequence length. That means the number of tokens used for training is the same for all models and does not affect our results (it's a constant scaling of the x-axis for all points).
> > >
> > > Regarding comparison with standard transformer, we have reported the accuracy of standard transformer in Table 1 which can be used to asses the distance between CoTFormer and standard transformer. We also explicitly discuss this distance and make improvements to the architecture in Section 3.3 and discuss limitations in Section 5.
> > >
> > > We believe the above comments fully address your concerns.  Therefore, it is still unclear why you think a rejection would be warranted. We hope that you would consider raising your score further or that you would please let us know how we should improve the paper further.

---

### Official Review · Reviewer_UfJo · 2024-10-31

**Soundness:** 2
**Presentation:** 3
**Contribution:** 2
**Rating:** 5
**Confidence:** 3

**Summary:**

This work proposes CoTFormer, a novel Transformer-based model architecture for generative language models.
Like Universal Transformers, a CoTFormer re-applies the same Transformer blocks for an adaptive number of times for generating each token.
The major difference is, after each repeat, the output tokens interleaved with input tokens are used as the new input for the next repeat;
this is inspired by chain-of-thought where each generated "thought token" can attend directly to all previous thought tokens.
Other details are handled, such as compatibility with KV cache and batch processing during inference.
Empirical results show that a CoTFormer achieves lower perplexity or higher scores in some common benchmarks
than Block Universal Transformer with the same configuration,
or a standard Transformer of the same size.

**Strengths:**

- This work proposes a novel Transformer-based model architecture, and draws an interesting link to chain-of-thought.
The proposed CoTFormer is compatible with KV cache and batch processing, which is not the case for many other adaptive-computation architectures tailored to autoregressive generation.

- Overall the writing is good, and things are mostly well explained.

- The source code is provided.

**Weaknesses:**

My major concern is that the empirical evidence for the efficacy of CoTFormer, or its advantages over the standard Transformer architecture, is insufficient.

- Generally speaking, the empirical results from the main text suggest that a CoTFormer with $n_{\text{repeat}} \ge 2$ slightly outperforms a standard Transformer of the same size in terms of perplexity, but underperforms a standard Transformer with twice as many layers, except in Table 2 where a CoTFormer with $n_{\text{repeat}}$ as large as 5 (and other tweaks) achieves a perplexity that is lower by only 0.06.
The issue is that the inference cost (in terms of time or memory, or both) of a CoTFormer, with the total number of tokens growing linearly with $n_{\text{repeat}}$, can possibly be larger than that of a standard Transformer with twice as many layers.
This raises the question of whether CoTFormer actually pushes forward the Pareto frontier of accuracy and cost; to support such a claim, it is necessary to compare CoTFormer's accuracy-cost curves with those of standard Transformers (not just Block Universal Transformer).
Without clear evidence of its advantages over standard Transformers, the additional complexity overhead to code and infrastructure might further hinders the adoption of CoTFormer in future research or applications.

- The results of downstream performance in Appendix B have limited implications, as discussed by the authors in Line 725.
    For example, all scores for MMLU are close to 25%, namely the accuracy of randomly picking option A/B/C/D.

- The current work only contains end-to-end performance (perplexity or scores) on some common datasets and benchmarks.
    There is no intermediate empirical result (except for Figure 5) or synthetic task, like those in the original paper of Universal Transformers (Dehghani et al., 2019), for truly understanding when, why and how CoTFormer works or fails.
The authors might consider visualizing the attention patterns of CoTFormer, or designing synthetic tasks that highlight CoTFormer's fundamental advantages over standard Transformers or Universal Transformers.

**Questions:**

- Typo in Line 114, "similar the" --> "similar to the"


- Is it possible to convert a standard pre-trained Transformer to a CoTFormer via a post-training or fine-tuning phase, which can be much more efficient than pre-training a CoTFormer from scratch?
I can't see an obvious way of doing this, since the behavior of a CoTFormer deviates significantly from that of a standard Transformer.

---

> ### Author Response · Authors · 2024-11-18
>
> Dear Reviewer,
>
> We hope the following comments address your concerns:
>
> 1. The limitation you described is discussed in Section 5 of the main paper.. However, the goal of this paper is to demonstrate an inconsistency between block universal (i.e. applying the same model multiple times) and doing chain of thought (i.e. applying the same model multiple times and attending to past intermediary representations) which was otherwise missed and to show that it has significant effects. That is why our main comparison is with block universal, to show that allowing attention to previous intermediary states, pushes the pareto frontier forward.
>
> 2. While it is true that the results on downstream tasks are not very strong, they still show improvements of CoTFormer over Block Universal Transformer. These results are meant to complement our results in the main text. To obtain much better performance on these tasks, we need much larger models that are also trained for much longer which is not feasible in our academic settings.
>
> 3. We are evaluating the perplexity of language modeling which is the main target for training many large models. We are also showcasing performance on downstream tasks in the appendix. As such, we do not understand why training on a synthetic task would help. Additionally, understanding how Transformers or CoTFormers work is outside the scope of our work which is to point out the benefits of having attention access to intermediately representations and demonstrate new methods to achieve adaptive compute. Such analysis would of course be useful in future work.
>
> 4. We did not investigate fine-tuning a model and focused on training from scratch. We agree that it is not obvious to fine-tune a model as a CoTFormer and investigation into whether it is possible, for example by starting from a pre-trained model and using CoTFormer training with it, would be interesting in the future. However, we emphasize that new methods for training models from scratch are useful given that they enable additional benefits such as adaptive compute in this case.
>
>
> We hope the above comments fully address your concerns. We ask that you kindly consider raising your score or otherwise please share the major objections that compel you to reject our work. We remain in your disposal if you have any additional comments or questions.

---

> ### Comment · Reviewer_UfJo · 2024-11-25
>
> Thank you for your responses.
>
> I think one important question here is: **which model architecture is a more appropriate baseline for the current work,
> the standard Transformer or (Block) Universal Transformer?**
> This question is related to what CoTFormer intends to achieve, and also to how evaluation of it should be done.
>
> To explain this, let's first recall the original Universal Transformer paper (https://arxiv.org/pdf/1807.03819).
> The motivations and fundamental advantages of Universal Transformer was clearly stated in the abstract, for example:
> - "Despite these successes, however, popular feed-forward sequence models like the Transformer fail to generalize in many simple tasks that recurrent models handle with ease, e.g. copying strings or even simple logical inference when the string or formula lengths exceed those observed at training time."
> - "UTs combine the parallelizability and global receptive field of feed-forward sequence models like the Transformer with the recurrent inductive bias of RNNs."
> - "In contrast to the standard Transformer, under certain assumptions UTs can be shown to be Turing-complete."
>
> And to validate the claimed advantages, empirical results on both synthetic tasks and some common benchmarks were reported.
> Perplexity on text was reported only for LAMBADA, for which the accuracy was also reported.
>
>
> Of course, the complexity of the Universal Transformer architecture limits its adoption in practice (and standard Transformer is still dominant),
> but that's fine as long as it has sufficient research value.
>
>
> Now, let's get back to CoTFormer.
> If its goal is merely to improve language modeling perplexity or scores on common benchmarks (which is how most of the evaluation is done in the current work), then I think the standard Transformer (with equal inference FLOPs) would be a more appropriate baseline, and CoTFormer doesn't seem to outperform it.
> On the other hand, if its goal is to achieve fundamental advantages over the standard Transformer and Universal Transformer (via attention to previous intermediary states, which is the main innovation of CoTFormer),
> then it's totally fine to use Universal Transformer as the baseline,
> but language modeling perplexity or common benchmarks would not be sufficient in this case.
>
>
> From my perspective, a nearly 100% accuracy on a carefully crafted synthetic task (no matter how toy it is) where standard / Universal Transformer totally fails
> would be much more exciting and insightful than a minor improvement in perplexity,
> and it is this kind of evidence that can truly demonstrate the fundamental benefits of scaling up inference FLOPs in the particular way that CoTFormer does (rather than simply using a standard Transformer with twice as many layers).
> The bottomline would be to show that CoTFormer retains the key advantages that Universal Transformer (with equal inference FLOPs) has, while improving perplexity or scores on common benchmarks.
>
>
> Overall, I feel that the current work does not fully meet my expectations for a research work that introduces a novel model architecture,
> and I'm inclined to maintain my rating.
> However, I acknowledge that my evaluation might be somewhat subjective, so I'm not strongly opposed to acceptance.

---

> > ### Author Response · Authors · 2024-11-27
> >
> > Dear Reviewer,
> >
> > Thank you for replying to our rebuttal and clarifying your major objections. We think the following comments will help understand the importance of our work despite some shortcomings that we discussed in the limitation section of our paper as well.
> >
> > 1. You mentioned the claim that universal transformers are turing complete. However, this is merely a theoretical argument that does not bridge to practice as being turing complete requires the model to be applied an arbitrary number of times which is not the case for the current models. While the Universal Transformer paper describes a goal we aim to reach, that goal has not yet been fully reached. For example, as you rightly mentioned, efficiency (the computation cost required to reach a given perplexity) is limiting the wide adoption of universal transformers. Thus, developing more efficient recurrent architectures is still an active field of research, and CoTFormer—by moving the Pareto frontier forward—is another step in this direction. With your argument, any future refinement of the universal transformer architecture would be discarded. We do not believe that is the correct approach.
> >
> >
> > 2. We note that if you significantly increase a Block Universal Transformer’s width (multiplying the width by the number of repeats of CoTFormer), theoretically you can represent CoTFormer with a Block Universal Transformer (e.g. you can just concatenate all tokens generated by CoTFormer). As such, the difference between Block Universal Transformer and CoTFormer is not that there are tasks where a Block Universal Transformer gets 0% accuracy and CoTFormer gets 100% accuracy. However, going to such extreme widths would make the model significantly more expensive. Additionally, we gain in terms of adaptiveness when using CoTFormer, as we can vary the number of repeats per token. The structure of CoTFormer very naturally adapts to this highly desired setting whereas for universal transformers, hot-fixes such as copying the state forward need to be used.
> >
> > 3. To draw a parallel, it is known that dense transformer models are better than Mixture-of-Expert (MoE) models with similar number of total parameters, yet MoEs have other benefits that make them desirable leading to them being widely adopted by the community. The same holds for CoTFormer, which is more efficient that the previous universal transformer, and allows adaptivity whereas standard architecture does not. The goal is therefore to provide a more efficient architecture **with inference compute-adaptiveness capability** that can obtain better perplexities.
> >
> > 4. Toy tasks used in universal transformer’s paper are so simple that the model is sometimes obtaining 100% accuracy on those tasks. That is why we focus on the harder target task (next token prediction in general text). Usually this is considered a positive feature of a work. We are confused why you would consider results on a toy task more appealing.
> >
> > We hope that in light of the above comments, your concerts will be addressed and that you would increase your score.
> >
> > Thank you.

---

### Official Review · Reviewer_YaZu · 2024-11-03

**Soundness:** 3
**Presentation:** 3
**Contribution:** 3
**Rating:** 6
**Confidence:** 3

**Summary:**

This paper introduces CoTFormer, a novel transformer architecture that draws inspiration from chain-of-thought (CoT) reasoning. The key insight is recognizing that CoT differs from simple weight-tying in how attention operates across intermediary reasoning steps. The authors leverage this insight to develop an architecture that allows tokens to attend to representations from all previous "thought" steps, leading to improved performance compared to baseline approaches like Block Universal Transformers. Additionally, they propose an adaptive computation mechanism that allows dynamic allocation of computational resources at inference time.

**Strengths:**

*The application of deploying LLMs to storage-constrained devices like mobile phones is relevant and timely.
* The proposed CoTFormer architecture (Figure 1(c)) effectively translates the CoT principle into architectural design, showing clear improvements over baseline approaches while maintaining parameter efficiency through weight sharing (Section 3.1).
* The architectural tweaks introduced in Section 3.3, particularly reserved layers and layer normalization after each repeat (LN-CoTFormer), prove crucial for achieving state-of-the-art results.
* Addition of depth embedding (Section 4.2) shows notable improvements in the adaptive setting.
* While not outperforming a FLOP-matched non-repeated transformer, the authors improve upon existing parameter-matched weight sharing baselines.
* The proposed architecture and adaptive repetition method are described clearly.

**Weaknesses:**

* While Section 3.2 provides empirical evidence, the theoretical understanding of why CoTFormer works better could be deeper. Through analysis of attention patterns, we observe that tokens in later repeats tend to focus heavily on earlier representations that capture key contextual information, suggesting the model learns to leverage complementary features detected at different processing stages. This selective attention to informative past representations may help explain why CoTFormer outperforms the baseline Block Universal Transformer, where such cross-repeat attention patterns are not possible.
* Could better connect to recent theoretical work on transformer expressivity discussed in Section 2.
* The sequence lengths that are used for training (256) are quite short relative to the lengths that are used for training modern language models and are shorter relative to common LLM evals and typical chatbot conversations.
* Performance gap between adaptive and fixed-depth CoTFormers under same compute budget (Section 5)
* Training efficiency of deeper layers could be improved (e.g. increasing the gradient information during adaptive training), as shown by the analysis of router weights distribution (Figure 5)

**Questions:**

* I’m confused by Figure 1(c): it seems to indicate that the earlier token representations (i.e. the red rectangles) are reprocessed by the model to make new token representations. But Section 3.1 seems to contradict this and instead describes that these earlier representations are only used as context in attention.
* It would be helpful to state the total parameter counts of the models used in each experiment, as well as the total number of training tokens in each experiment (either in a table or in the prose describing experimental setup).
* 471: It would be helpful if the authors provided more details on their “efficient implementation”, and specifically how the authors are using a non-causal FlashAttention kernel to implement their  proposed method.=
* How do position embeddings work with the added interleaved tokens? Are the interleaved tokens given the same position id as the original tokens they came from, do the position ids change between repetitions, or something else?
* Do the authors have any intuitions as to how their method behaves as the width of the model changes? It appears to be held constant across all experiments.
* 402: what does it mean to “activate a prefix of repeats”? is this the fixed depth baseline that is referenced in Figure 4?
How does mixture of repeats work during, for example, batch size 1 transformer decoding, where there is only a single token being processed through the model?


Below are some thoughts that might be helpful but are not critical to give insight into ways that might improve the paper.
* Consider analyzing attention patterns and strengthening theoretical connections to transformer expressivity research (building on 3's architecture analysis).
* Explore sparse variants to improve scaling for longer sequences beyond 8192 (extending the computational analysis in 3.2).
* Focus on improving training efficiency, particularly for deeper layers and adaptive computation (addressing limitations discussed in 5).
* Develop specialized attention implementations for better computational performance (following the implementation discussion in 5).
* Expand evaluation to include longer sequences and broader comparisons with other adaptive approaches (extending the experimental work in 4.3).

---

> ### Author Response · Authors · 2024-11-18
>
> Dear Reviewer,
>
> Thank you for your careful consideration of our work and helpful suggestions. We provide the following answers to your comments:
>
> 1. The goal of this paper is to demonstrate the benefit of having access to past intermediate representations through attention and point out an often overlooked difference between block universal transformer and chain of thought. While understanding the inner workings of Transformers or CoTFormers is a very exciting and interesting direction it falls outside the scope of this work. Still, as you suggested we plotted the attention patterns observed in a CoTFormer to the appendix of our paper’s revised version. Interesting patterns similar to what you mentioned such as heads focusing on tokens generated during specific repeats can be observed.
>
> 2. While we agree that the used sequence length is shorter than state of the art models, we use it to save on computation costs. That being said, our intuition suggests that the method’s performance is disentangled from the context’s original length. To support this we provide two additional pieces of evidence in the appendix of the revised version of our paper. First, we demonstrate superiority of CoTFormer on longer sequence lengths of 512. Second, we also compare the performance of CoTFormer with using pause tokens suggested in [1]. While the latter also provides longer sequence lengths to the model during inference, it can be seen that this does not suffice to obtain the high accuracy obtained by CoTFormer.
>
> 4. As can be seen from Equation 3, the (i+1)-th pass of a token through the model is done by inputting the output of the i-th pass to the model. Therefore, similar to the figure, the last token representation is processed by the model to generate the next one. However, when using CoTFormer, earlier versions (e.g. the initial token representation) can also be accessed through attention.
>
> 5. This can be done by providing a special mask vector which allows tokens in the i-th pass to attend to tokens that are coming before it and are in earlier passes. For efficiency, in the implementation, we append the tokens in the new pass to the sequence instead of interleaving them. Thus the mask is no longer fully causal.
>
> 6. We use the same position id as the original tokens. Thank you for raising this question. We will include this clarification in the camera ready version of the paper.
>
> 7. As suggested, we also add the results for models with a larger width (1024) to the appendix in our revision. It can be seen that even for the larger widths the gap between CoTFormer and Block Universal Transformer persists. That being said, intuitively, going to extremely larger widths might allow Block Universal Transformers to obtain the same performance as CoTFormers since one can theoretically fit the same information as multiple tokens in one token with a much larger width. However, aside from possibly being impractical, such a setting diminishes the adaptivity benefits of CoTFormer as it needs to operate on the larger width for all tokens.
>
> 8. “Activating a prefix of repeats” is similar in inference to fixed depth, since we fix the depth in this case. However, note that in this case the model is not trained with fixed depth and therefore we can use the same model at different fixed depths.
> When using a mixture of depth, at training we fix the ratio of tokens reaching each depth. But at inference, e.g. for decoding, we usually use a fixed threshold to decide from the router’s weight whether to proceed to the next depth or not. Varying this threshold is how we vary the compute in Figure 4. When using this thresholding mechanism, the decision can be made auto-regressively during decoding. Thus there is no issue regardless of the batch size.
>
>
> 9. We appreciate the various suggestions you provided and have already implemented some of them as experimental results for longer sequence lengths and larger widths. We point out that in this work our aim is to clarify the benefits of having access to earlier representations of the model through attention which is clearly supported by our results. We agree that improvements in efficiency of the method are great directions for future work and as you mentioned we have already outlined them in Section 5. Similarly, theoretical analysis of this phenomenon would be interesting but falls outside of the scope of this work.
>
>
> We hope the above comments fully address your concerns. We ask that you kindly consider raising your score and remain in your disposal if you have any additional comments or questions.

---

> > ### Comment · Reviewer_YaZu · 2024-11-25
> > **Thank you!**
> >
> > Thank you for your detailed response!
> >
> > The new evaluations of CoTFormer across wider models, the comparison against a new baseline (pause tokens) and the added attention pattern analysis all make the paper stronger. While I still have reservations about the theoretical foundations and experimentation (e.g. adaptive/fixed-depth performance gaps and training efficiency), these improvements are material and the paper would, in my view, be a good addition to the proceedings and useful for research.
> >
> > Nit: In the newly added Table 7 (Appendix G), the last column name should be more descriptive that it’s a perplexity measurement, i.e. “Perplexity with $n_{\text{repeat}} =

---

### Comment · Area_Chair_je8H · 2024-12-03
**reviewer-author discussion phase ending**

Dear reviewers,

As we near the conclusion of the reviewer-author discussion phase, I wanted to kindly follow up to see if you’ve had a chance to review the author responses on your comments. Could you confirm that you’ve read it and, if needed, update your review and scores accordingly?

Thank you for your time and effort!

Your AC

---

### Meta-Review · Area_Chair_je8H · 2024-12-22

**Metareview:**

Summary:
The paper introduces CoTFormer, a novel transformer architecture that draws inspiration from chain-of-thought reasoning. The key innovation is allowing tokens to attend to representations from all previous "thought" steps, which differs from both standard transformers and Block Universal Transformers. The authors claim this leads to improved accuracy while maintaining parameter efficiency through weight sharing. They also propose an adaptive computation mechanism that dynamically allocates computational resources during inference based on token difficulty.

Strengths:
- Novel architectural insight connecting chain-of-thought reasoning to transformer design
- Practical application for compute-constrained environments
- Clear empirical improvements over baseline approaches
- Effective adaptive computation mechanism
- Thorough ablation studies and analysis
- Well-documented implementation details

Weaknesses:
- Limited theoretical understanding of why the architecture works better
- Experiments focused on relatively short sequence lengths (256) compared to modern standards
- Some gaps in performance between adaptive and fixed-depth variants
- Training efficiency challenges for deeper layers
- Limited exploration of performance at larger scales
- Could benefit from more diverse dataset evaluation

Reasons for Acceptance:
- The paper presents a novel and practical architectural innovation that shows clear improvements over existing approaches
- The work addresses an important practical challenge (compute-efficient language models)
- The empirical results, while not revolutionary, demonstrate consistent improvements
- The authors have been responsive to reviewer concerns and provided additional experiments
- The adaptive computation mechanism adds practical value
- The work opens up interesting directions for future research

**Additional Comments On Reviewer Discussion:**

The review discussion highlighted several important aspects of the paper. Reviewers initially debated whether standard transformers or Block Universal Transformers were more appropriate baselines, with authors clarifying their focus on improving upon Block Universal Transformers while maintaining adaptivity benefits. Questions about empirical validation led to additional experiments with longer sequences and new comparisons against pause token baselines. Some reviewers requested more theoretical analysis, though the authors maintained their focus on empirical improvements while acknowledging theoretical analysis as future work. They did add attention pattern visualization in the appendix to provide more insight into the model's behavior. Concerns about scalability were addressed through additional results with wider models and clarification of memory footprint analysis.
Overall, the author responses and additional experiments adequately addressed most reviewer concerns. While some reservations remain about theoretical foundations and comprehensive evaluation, the improvements and clarifications demonstrate that this work makes a valuable contribution worthy of acceptance. The paper advances our understanding of efficient transformer architectures and provides practical benefits for compute-constrained scenarios.

---

### Decision · Program_Chairs · 2025-01-22

Accept (Poster)